# Probing the causal involvement of dlPFC in directed forgetting using rTMS—A replication study

Benjamin J. Stauch[1,2,3], Verena Braun[3], Simon Hanslmayr[3]*

**1** Ernst Strüngmann Institute (ESI) for Neuroscience in Cooperation with Max Planck Society, Frankfurt, Germany, **2** International Max Planck Research School for Neural Circuits, Frankfurt, Germany, **3** School of Psychology, University of Birmingham, Edgbaston, Birmingham, United Kingdom

* s.hanslmayr@bham.ac.uk

**Data Availability Statement:** Preregistration, standardized procedure, analysis scripts and directed forgetting and Stroop result datasets can be retrieved from https://osf.io/esh69/.

**Funding:** SH was supported by the European Research Council (grant agreement N°647954,

## Abstract

The forgetting of previously remembered information has, for a long time, been explained by purely passive processes. This viewpoint has been challenged by the finding that humans show worse memory for specific items that they have been instructed to forget. The dorsolateral prefrontal cortex has, through imaging, lesion and brain stimulation studies, been implied in controlling such active forgetting processes. In this study, we attempted to solidify evidence for such a causal role of the dlPFC in directed forgetting by replicating an existing rTMS study (Hanslmayr S, 2012) in a preregistered within-participant design. We stimulated participants at the dlPFC (BA9) or vertex using 45s of 1Hz rTMS after instructions to forget previously remembered words in a list-method directed forgetting paradigm and tested for effects on the amount of forgotten information. Contrary to the study we were attempting to replicate, no significant increase in forgetting under dlPFC stimulation was found in our participants. However, when combining our results with the study we were attempting to replicate, dlPFC stimulation led to significantly increased directed forgetting in both studies combined. We further explored if the rTMS parameters used here and in earlier work (Hanslmayr S, 2012) influenced inhibitory processing at their time of delivery or in a more persistent manner. Unaltered incongruency and negative priming effects in a Stroop task conducted directly after stimulation suggests that our rTMS stimulation did not continue to influence inhibitory processing after the time of stimulation. As the combined evidence for increased directed forgetting due to rTMS dlPFC stimulation is still quite weak, additional replications are necessary to show that directed forgetting is indeed causally driven by an active prefrontal process.

## Introduction

In general, forgetting is mostly seen as a passive process. Nevertheless, people are also able to specifically forget outdated or unwanted information—a phenomenon known as directed forgetting [1].

https://erc.europa.eu), the Economic and Social Research Council (ESRC grant agreement N°ES/R010072/1, https://esrc.ukri.org), and Deutsche Forschungsgemeinschaft (HA 5622/1-1, https://www.dfg.de/). The funders had no role in study design, data collection and analysis, decision to publish, or preparation of the paper.

**Competing interests:** The authors have declared that no competing interests exist.

Robert Bjork defined directed forgetting as "situations in which (a) there has been a prior attempt, however brief or extended, to learn the material that is now to be forgotten and (b) there is an explicit (or totally unambiguous implicit) cue to forget that material." [2, p. 462]. Directed forgetting effects have been shown in a wide variety of tasks using diverse stimuli, such as words, images, and autobiographic memories [3]. In nearly all directed forgetting paradigms, participants are informed that their task is to remember some information presented to them (usually words) for a later memory test. They are then presented with this information. Immediately following either each item (item-method directed forgetting, IDF) or a list of items (list-method directed forgetting, LDF), participants are instructed to either remember or forget the information they were just shown. Contrary to instructions, both to-be-remembered (TBR) and to-be-forgotten (TBF) items are tested at a later time. Usually, two effects of instructions to forget emerge: Firstly, to-be-forgotten items are remembered less than to-be-remembered ones (the forgetting effect). Secondly, to-be-remembered items are remembered better when being presented with or after to-be-forgotten items compared to additional to-be-remembered ones (enhancement effect). For an overview of a list-method directed forgetting paradigm as used in this study, see Fig 1A and 1B.

In both item-method and list-method directed forgetting tasks, participants consistently show worse memory performance for to-be-forgotten items in free recall tests [4–6]. Both attempting to forget as well as successfully forgetting previously learned information has been repeatedly linked to increased activity in the dorsolateral prefrontal cortex (dlPFC) using both fMRI [7, 8, 9] and EEG [10]. This increased dlPFC activity has been connected to decreased long-range synchrony in the alpha band [7] and decreased hippocampal activity [9, 11]. As the dlPFC has been understood to play a crucial role in intrinsic inhibitory control [12] and

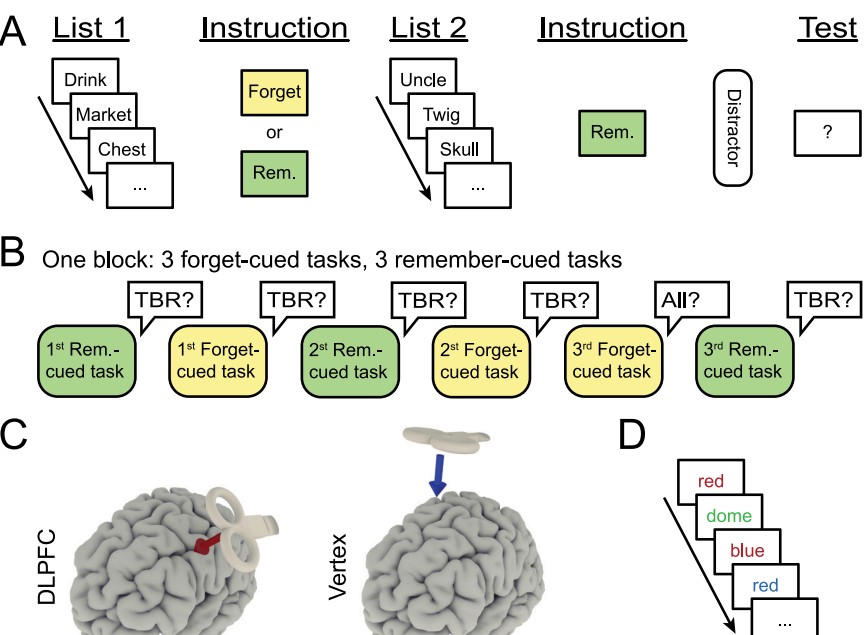

**Fig 1. Experiment design.** A: A list-method directed forgetting paradigm followed by a free recall test, as used in this study. B: An example of one stimulation condition block, consisting of six list-method directed forgetting tasks in pseudorandom order. Participants were instructed to recall all words (both to-be-remembered and to-be-forgotten) only after the last forget-cued task per block. TBR = to-be-remembered. Diagrams adapted from Hanslmayr et al. (2012). C: Approximate stimulation locations, displayed on an example brain surface. D: A computerized serial Stroop task, as used in this study. Congruent, neutral, incongruent, and negative priming trials were shown.

neuronal synchrony in the alpha-beta band has been mapped to top-down processing [13] and successful item retention in working memory tasks [14], it has been postulated that directed forgetting can be explained by a prefrontal control system that "can be targeted flexibly at different stages of mnemonic processing and at different types of representation to modulate the state of traces in memory" [3].

The lateralization of frontal brain activation in directed forgetting tasks is still an active matter of research. In item-method directed forgetting paradigms, both right-lateralized [9] and bilateral [8] activation of frontal areas has been found. In list-method directed forgetting paradigms, left-lateralized frontal activation was found [7]. To ensure that we stimulated an area directly implicated in a task highly similar to ours, we chose to stimulate at the location of strongest activation found in the study we were attempting to replicate [7].

In order to test the causal role of dlPFC in directed forgetting, a series of lesion and brain stimulation studies have been conducted. Subjects with frontal brain lesions showed impaired directed forgetting, meaning full memory performance for to-be-forgotten items, in both item-method and list-method directed forgetting tasks compared to subjects with parietal lesions and healthy controls [15]. In a tDCS study, inhibitory stimulation of the dlPFC (anodal stimulation over left dlPFC, cathodal over right dlPFC) prevented directed forgetting [16]. The rTMS study we are attempting to replicate stimulated left dlPFC (specifically BA9) after the forget or remember instruction using 45 seconds of 1Hz rTMS. Stimulating the dlPFC caused participants to forget even more to-be-forgotten items than stimulation at a control site (vertex) [7].

As can be seen, activity in prefrontal cortex, especially dlPFC, has been shown to be both brought about by and necessary for forgetting in directed forgetting tasks. It should be noted, however, that all lesion or stimulation studies conducted so far used between-subject designs: Participants with frontal lesions or under stimulation were compared to healthy participants or participants undergoing some kind of sham stimulation. This hinders the interpretability of the results, as the lesion or stimulation might induce side effects unrelated to dlPFC functioning. For example, stimulation might (compared to sham stimulation) influence cortical processing in a general way, such as by increasing tiredness or distractibility. In order to control for such between-group confounding effects, we decided to replicate the 1Hz rTMS list-method directed forgetting paradigm used in this previous study [7] in a within-subject rTMS design.

Hanslmayr et al. used rTMS parameters not directly comparable to other studies applying dlPFC stimulation (short-term, online rTMS instead of several minutes of stimulation before the task [7]). It is unclear if these pulses had an effect directly at their time of delivery (i.e. during the remember/forget cue) or if their effect was more cumulative and long-term. To test if the applied rTMS parameters influenced inhibitory processing not only during, but also after their application, we ran a computerized Stroop task with incongruent and negative priming trials after every stimulation condition block. If the dlPFC stimulation had an accumulating and persisting effect on inhibitory processing, reaction times in incongruent and negative priming trials should be affected by the stimulation site during the preceding task block.

To our knowledge, this study is the first to run two list-wise directed forgetting paradigms in succession with the same subjects. As the test instruction after each directed forgetting task asks participants to produce the words they were instructed to forget, one might expect them to not follow the forget instruction in their second task. We developed a simple cover story to minimize this possibility. In order to check whether participants followed the forget instruction in the second task or whether running the list-method directed forgetting task twice diminished effects of instructions to forget, we checked for decreased forgetting effects in the second task.

## Materials and methods

### Preregistration and data availability

Experiment design, target sample size, exclusion criteria as well as the statistical tests presented under Preregistered analyses were preregistered before data collection. Preregistration, standardized procedure, analysis scripts and directed forgetting and Stroop result datasets can be retrieved from https://osf.io/esh69/.

### Participants

We analyzed 24 right-handed participants (mean age = 19.2, range 18-28; 4 males) from the Birmingham University Psychology student sample pool, which gave us 95% power to detect an effect of the same size as found in [7]. Six further participants were tested but excluded due to our preregistered exclusion criteria. Participants were native English speakers, had never been diagnosed with any neurological or psychiatric disorders and did not present any contraindications against TMS and MRI application. All participants gave written informed consent. The protocol was approved by the ethics committee of the University of Birmingham.

### Preregistered exclusion criteria

We excluded any participant showing forgetting effects more than 2.5 median absolute deviations (MADs) from the sample median [17] in at least one condition (n = 4). MAD is a robust measure of dispersion and is calculated as $MAD = bM(x_i - M_j(x_j))$, meaning $b$ times the median of the deviations of all observations from the sample median, where $b$ = 1.4826 when a normal distribution is assumed in the sampling population [17]. We also excluded any participant who answered yes to the question "During the experiment, did you actively rehearse the words you were instructed to forget?" in our post-experiment questionnaire (n = 2). Exclusions were done when 24 participants were collected, after which we re-recruited up to the target sample size.

### Directed forgetting paradigm

As study material, 240 words were drawn from the MRC Psycholinguistic Database [18] and split into 24 ten-word lists. The lists were matched for number of letters ($M$ = 5.32, $SD$ = 1.45), number of syllables ($M$ = 1.51, $SD$ = 0.66), concreteness ($M$ = 543.08, $SD$ = 45.07), imaginability ($M$ = 561.47, $SD$ = 38.19) and word frequency ($M$ = 58.96, $SD$ = 82.67). The lists used in our study and the lists used in [7] did not differ in any of these characteristics (all $p$ > 0.5). Between participants, each of the 24 lists was used equally often in each condition—list 1 before a remember cue (R1), list 1 before a forget cue (F1), list 2 after a remember cue (R2), and list 2 after a forget cue (F2).

For each task, two ten-word lists were shown (Fig 1A). The words of each list were presented sequentially. Before each word, a central fixation cross was shown for a random duration between 1.5s and 2.5s. Each word was shown for 2.5s, written in black on a gray background. Between the two lists of each directed forgetting task, participants were shown a cue to remember or forget the just-presented words of list 1 for 5s. The second list of each task was always followed by a cue to remember, also shown for 5s. After the second cue, participants were instructed to count down verbally from a randomly chosen number between 300 and 1000 in steps of 3 for two minutes as a distractor task. After the distractor, participants were instructed to recall, in any order they preferred, any words they were previously cued to remember (R1 and R2 in remember-cued tasks, F2 in forget-cued tasks).

Subjects completed six list-wise directed forgetting tasks per stimulation condition block (dlPFC or vertex, Fig 1B). The order of forget-cued and remember-cued tasks was pseudorandomized over the six tasks, such that the first, second and third task pair contained one forget-cued and one remember-cued task each. After the forget-cued task of the third task pair of each stimulation condition block, participants were informed that the forget cue had been shown due to a programming error and were asked to recall words from both the to-be-forgotten (F1) as well as the to-be-remembered list (F2). Only memory performance in the last task pair of each stimulation condition block was analyzed. Memory performance in those tasks will be referred to as F1 and F2 for list 1 and 2 of the forget-cued and as R1 and R2 for list 1 and 2 of the remember-cued task in both stimulation conditions.

## TMS

During each list 2 of the twelve directed forgetting paradigm, 45 pulses of 1Hz rTMS at 90% of individual resting motor threshold were delivered to either left dlPFC (MNI coordinates: $x = -45$, $y = 6$, $z = 39$) or vertex (MNI coordinates: $x = 0$, $y = -10$, $z = 80$), the same coordinates used in [7]. Stimulation target was varied block-wise, whereby each block consisted of half of the directed forgetting tasks. Block order was counterbalanced between participants. Due to the varied intertrial interval, word presentation and TMS pulses did not co-occur in any regular manner. For TMS delivery, a Magstim Rapid stimulator with a figure-of-eight-shaped coil was used. To achieve precise TMS targeting, pulse delivery was guided by the Brainsight Neuronavigation participant 3D tracking system using individual MRI scans. T1-weighted high resolution ($1mm^3$) brain scans were acquired for each participant using a 3T Philips Achieva MRI scanner before the experiment. Using the twelve-parameter linear transform function implemented in FLIRT [19], a participant-to-MNI152 transformation matrix was created, inverted, and then used to transform the MNI target coordinates into participant-specific target overlays. These were then used to guide TMS delivery during the experiment itself.

## Stroop paradigm

Before the experiment (as training) as well as after each stimulation condition block, the participants undertook a Stroop task (Fig 1C). The task consisted of four different trial types: congruent, neutral, incongruent, and negative priming trials. For congruent trials, the words "red", "green" or "blue" were shown in their respective font color (red, green, and blue). For neutral trials, three non-color words of similar imaginability, concreteness, syllable number, letter number, valence and arousal ratings to red, green and blue were drawn from the MRC Psycholinguistic Database [18]: "dome", "hop" and "goal". They were also presented in either red, green or blue font color. For the incongruent trials, the words "red", "green" or "blue" were presented in red, green or blue font color such that color word and font color differed (for example, the word "green" presented in a red font). Negative priming trials always followed incongruent trials and also had non-matching color word and font color, whereby their font color was identical to the color word of the just-presented incongruent stimulus (for example, if the incongruent stimulus was "blue" presented in a red font, the following negative priming stimulus could have been "red" or "green" presented in a blue font). For each Stroop task, 200 trials (50 per trial type) were presented in a pseudorandomized order: An incongruent trial was always followed by a negative priming trial, and each trial type appeared once each four trials. Participants were instructed to ignore the words and to name the font color of each stimulus using the arrow keys of a USB keyboard, which had been marked with colored stickers. Stimuli were presented on a black background until a response was made. The intertrial interval was 1000ms and consisted of the presentation of a central fixation cross.

## Procedure

Upon arrival, participants were informed that they were taking part in a memory experiment. Before the start of the first directed forgetting block, two practice list-wise directed forgetting tasks, one remember-cued and one forget-cued, were conducted, during which verbal feedback was given. In addition, a practice Stroop task of 200 trials was run. After the practice tasks, one of the two stimulation condition directed forgetting blocks was run, followed by the first analyzed Stroop task—which took around five minutes—and a five minute break. After the break, the second stimulation condition directed forgetting blocks was run, followed by the second analyzed Stroop task. During the experiment, two experimenters were in the room with the participant: One to deliver rTMS stimulation, the other one to give verbal instructions and to manually record memory performance. During the Stroop tasks, the experimenters left the room.

## Preregistered analyses

Memory performance was only analyzed for the last block pairs (remember and forget condition) of each stimulation condition. These contained the forget block in which participants were asked to report the to-be-forgotten words.

The forgetting effect was calculated per participant as the difference between percentage of List 1 words recalled following a remember or forget cue (R1—F1). The enhancement effect was calculated per participant as the difference between percentage of List 2 words recalled following a forget or remember cue (F2—R2). To test for effects of dlPFC vs. vertex rTMS stimulation on forgetting and enhancement scores, we calculated paired t-tests to replicate the tests done in Hanslmayr et al. [7]. In addition, we also compared forgetting and enhancement scores between the stimulation conditions using exact Wilcoxon-Pratt signed-rank tests, as we expected our results (percentage of words remembered) to be non-normally distributed. In order to quantify evidence for no effect, we computed $BF_{01}$ values using a Bayesian t-test with a non-informative Cauchy prior, prior width = 0.707, for all preregistered non-significant t-tests [20]. $BF_{01}$ can be interpreted as the likelihood of the data given $H_0$ divided by the likelihood of the data given $H_1$. The higher $BF_{01}$, the more likely is the data given $H_0$ compared to the data given $H_1$ [21].

As dlPFC stimulation was found to increase forgetting effects in the study we were attempting to replicate [7], these tests and values were computed one-tailed for forgetting effects. All other tests were computed two-tailed. The alpha level was set to $\alpha = 0.05$. Hedge's $g$ with 95% confidence intervals [22] was reported as an effect size. To compare the effect of dlPFC rTMS stimulation on forgetting versus enhancement scores, we computed a repeated measures ANOVA [23] with the within-participant factors stimulation target (dlPFC vs. vertex) and score type (forgetting vs. enhancement).

## Exploratory analyses

To assess if list output order was modified by dlPFC stimulation, we computed linear regressions of cue (remember vs. forget), stimulation site (dlPFC vs. vertex) and the interaction of these factors on the word list that participants recalled as their reported words 1-5.

To integrate our results with existing results [7], we combined the results from the original study and our replication attempt in a continuously cumulating meta-analysis [24]. To do so, we ran a weighted fixed-effect meta-analysis [25] for forgetting and enhancement effect modulations due to dlPFC stimulation in the two studies.

In all other exploratory analyses, Welch's t-tests or paired t-tests were used for normally distributed data, while non-normal data were compared using the exact Wilcoxon-Mann-

Whitney test. Binomial ratios (such as the gender ratios between our study and [7]) were compared using Pearson's Chi-squared test. For the Stroop task analysis, we removed any trials with reaction times more than 2.5 MADs from participant cell median reaction time. We then analyzed participant cell mean reaction times using a a repeated measures ANOVA (Greenhouse-Geisser-corrected where appropriate) with the within-participant factors trial type (congruent vs. neutral vs. incongruent vs. negative priming) and pre-Stroop stimulation target (dlPFC vs. vertex). We only analyzed reaction times from correct trials.

# Results

## Manipulation checks

Our paradigm induced a forgetting effect: Participants recalled less list 1 words when they were told to forget list 1 than when they were told to remember list 1 under both vertex ($t(23)$ = 3.22, $p$ = 0.004, $g$ = 0.57 [0.20, 0.95]) and dlPFC stimulation ($t(23)$ = 4.28, $p$ = 0.0003, $g$ = 1.01 [0.44, 1.59]). Furthermore, our paradigm also managed to induce an enhancement effect: Participants recalled more list 2 words when they were told to forget list 1 than when they were told to remember list 1 under both vertex ($t(23)$ = 2.72, $p$ = 0.01, $g$ = 0.72 [0.13, 1.31]) and dlPFC stimulation ($t(23)$ = 2.28, $p$ = 0.03, $g$ = 0.38 [0.04, 0.72]).

## Preregistered analyses

Contrary to our prediction, participants did not show a significant increase in forgetting (Fig 2A) under dlPFC stimulation ($M$ = 27.1%$pt$, $SD$ = 31.0%$pt$) compared to vertex stimulation ($M$ = 17.1%$pt$, $SD$ = 26.0%$pt$, $r$ = −0.18) in the t-test ($t(23)$ = 1.12, $p_{one-sided}$ = 0.14, $g$ = 0.34 [−0.27, 0.95], $BF_{01}$ = 1.57) or the signed-rank test ($Z$ = 1.03, $p_{one-sided}$ = 0.16). The $BF_{01}$ = 1.57 can be interpreted as meaning that the data were only 1.57 times more likely to be measured if dlPFC stimulation (compared to vertex stimulation) had no effect on forgetting than if dlPFC stimulation (compared to vertex stimulation) increased forgetting.

As expected, participants did not show a significantly different enhancement score (Fig 2B) under dlPFC stimulation ($M$ = 11.3%, $SD$ = 24.2%) compared to vertex stimulation ($M$ = 18.3%, $SD$ = 33.1%) in either the t-test ($t(23)$ = 0.87, $p$ = 0.39, $g$ = −0.24 [−0.78, 0.30], $BF_{01}$ = 3.31) or the signed-rank test ($Z$ = −0.69, $p$ = 0.50). The $BF_{01}$ = 3.31 can be interpreted as meaning that the data were 3.31 times more likely to be measured if dlPFC stimulation

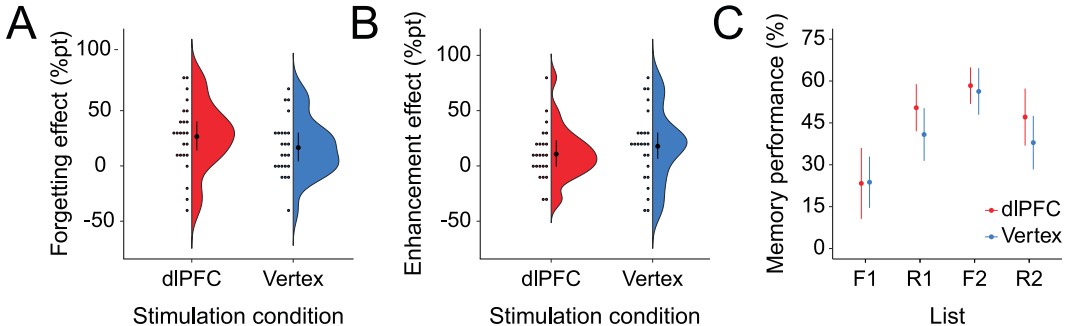

**Fig 2. No statistically significant effect of stimulation site on forgetting and enhancement effects.** A: Mean forgetting effect in percentage points (memory performance in list R1—memory performance in list L1) under dlPFC and vertex stimulation. B: Mean enhancement effect in percentage points (memory performance in list F2—memory performance in list R1) under dlPFC and vertex stimulation. Colored area represents effect distribution over the 24 participants as a Gaussian kernel density plot. C: Mean memory performance in all lists and conditions—list 1 before a remember cue (R1), list 1 before a forget cue (F1), list 2 after a remember cue (R2), and list 2 after a forget cue (F2). Error bars represent within-participant 95% CIs in all plots [26].

(compared to vertex stimulation) had no effect on enhancement than if dlPFC stimulation (compared to vertex stimulation) influenced enhancement in either direction.

A two-way repeated measures ANOVA with the factors stimulation (vertex vs. dlPFC) and effect type (forgetting vs. enhancement) did not show the predicted interaction between stimulation and effect type ($F1, 23$) = 1.72, $p$ = 0.20, Fig 2C). Forgetting and enhancement effects were not differently influenced by dlPFC versus vertex stimulation to a significant amount.

## Exploratory analyses

As the enhancement effect depends on output order [27], we checked if dlPFC stimulation modified output order. While subjects were more likely to start recall with list 1 when previously instructed to remember list 1 than when previously instructed to forget it ($p < 0.05$ for recalled words 1-4), stimulation site did not influence recall order (all $p > 0.05$ for recalled words 1-5) nor moderate the difference in recall order between the remember and forget condition (all $p > 0.05$ for words 1-5).

Results from the Stroop task implied that the rTMS parameters used in our study and in [7] did not continue to influence inhibitory processing after their application (Fig 3). The incongruency effect (incongruent—congruent trials) did not differ between the Stroop tests conducted after vertex ($M = 32ms$, $SD = 32ms$) vs. after dlPFC stimulation ($M = 39ms$, $SD = 39ms$, $r = -0.23$, $t(23) = 0.66$, $p = 0.52$, $g = 0.20$ [−0.41, 0.82], $BF_{01} = 3.82$). The same pattern held when the incongruency effect was computed with neutral trials as baseline: it did not differ between Stroop tests after vertex ($M = 19ms$, $SD = 61ms$) vs. after dlPFC stimulation ($M = 35ms$, $SD = 37ms$, $r = 0.17$, $t(23) = 1.21$, $p = 0.24$, $g = 0.30$ [−0.20, 0.82], $BF_{01} = 2.44$).

In addition, negative priming effects (negative priming trials—incongruent trials) did not differ between the Stroop task conducted after vertex stimulation ($M = 38ms$, $SD = 45ms$) vs.

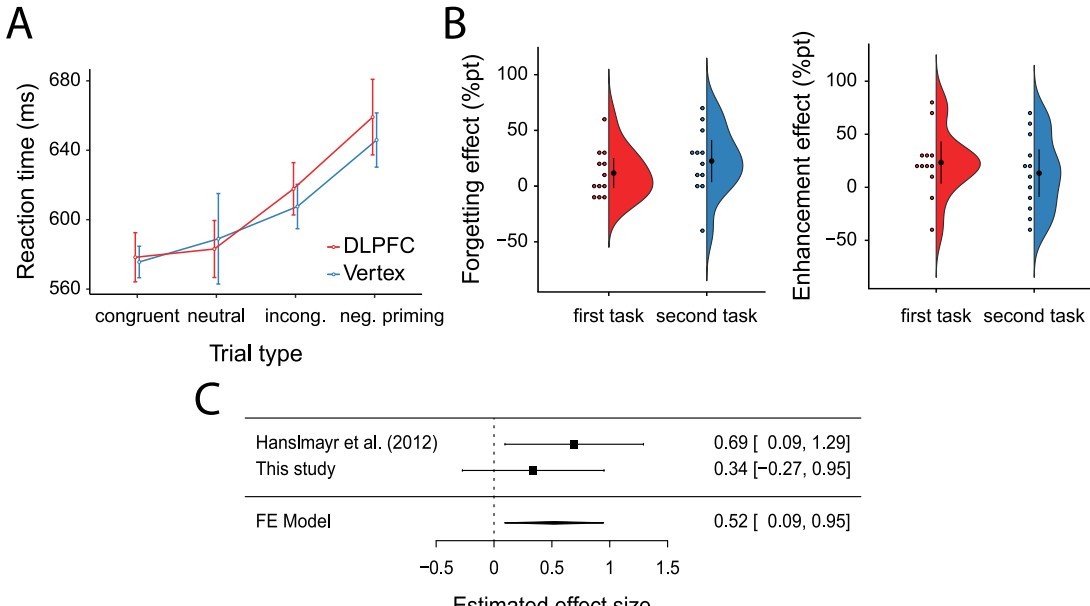

**Fig 3. Exploratory analyses.** A: Mean reaction times in ms in all trial types of the Stroop task after dlPFC or vertex stimulation. Error bars represent within-participant 95% CIs [26]. B: Forgetting and enhancement effect under vertex stimulation during the first task block vs. the second task block over participants. Error bars represent 95% CIs. C: Forest plot of a fixed effect meta-analytical combination of the two studies conducted on the effect of dlPFC 1Hz rTMS stimulation on forgetting in list-method directed forgetting so far. Error bars represent 95% confidence intervals, effect size is Hedge's g.

after dlPFC stimulation ($M = 41ms$, $SD = 61ms$, $r = -0.10$, $t(23) = 0.19$, $p = 0.85$, $g = 0.05$ [−0.52, 0.63], $BF_{01} = 4.59$).

Over participants, the difference between the forgetting effect under dlPFC vs. under vertex stimulation correlated weakly with the difference between the incongruency effect after dlPFC stimulation vs. after vertex stimulation ($r = 0.43$ [0.040.71], $p = 0.03$), but not with the difference between the negative priming effect after dlPFC stimulation vs. after vertex stimulation ($r = -0.23$ [−0.58, 0.19], $p = 0.28$).

While trial type (congruent/neutral/incongruent/negative priming) had a significant main effect on reaction times in a repeated measures ANOVA ($F(1.67, 38.5) = 31.14$, $p < 0.0001$, $\eta^2_G = 0.09$), pre-Stroop stimulation site (dlPFC/vertex) had not $F(1, 23) = 0.49$, $p = 0.49$, $\eta^2_G = 0.0007$. Importantly, there was no significant interaction between trial type and stimulation site ($F(2.17, 49.81) = 0.71$, $p = 0.51$, $\eta^2_G = 0.001$). All trial types had similar effects under both pre-Stroop dlPFC stimulation conditions (see Fig 3A), fitting the idea that no specific aftereffects on inhibitory processing persisted after stimulation.

## Effects of running a list-method directed forgetting paradigm twice

As hoped, there was no significant difference between the forgetting effect (under vertex stimulation) between the first ($M = 11.67\%$, $SD = 21.25\%$) and the second analyzed directed forgetting task ($M = 22.50\%$, $SD = 29.89\%$, $t(19.86) = 1.02$, $p = 0.32$, $g = 0.40$ [−0.38, 1.18]). Also as hoped, there was no significant difference between the enhancement effect (under vertex stimulation) between the first ($M = 23.33\%$, $SD = 31.43\%$) and the second analyzed directed forgetting task ($M = 13.33\%$, $SD = 35.25\%$, $t(21.72) = 0.73$, $p = 0.47$, $g = -0.29$ [−1.07, 0.49]). The effects of our paradigm were not significantly influenced by being measured a second time after instructing participants to recall to-be-forgotten items at the end of the first directed forgetting block. In a debriefing conversation after the experiment, none of our analyzed participants reported ignoring the forget instruction in the second directed forgetting block (after having been asked to recall to-be-forgotten words), while both excluded participants reported ignoring the forget instruction throughout the whole experiment. These results show that it is feasible to repeat the list method directed forgetting paradigm at least twice in the same subjects without impairing the amount of forgetting.

## Integration with Hanslmayr et al. (2012)

In order to integrate our finding of non-significant boosting of forgetting due to dlPFC stimulation ($g = 0.34$ [−0.27, 0.95]) with the findings we were attempting to replicate [7], we computed an effect size from their original data ($g = 0.69$ [0.09, 1.29]) and combined the effects from both studies using a fixed-effect meta-analysis. Combining both studies, rTMS stimulation of left dlPFC significantly boosted forgetting ($g = 0.52$ [0.09, 0.95], $Z = 2.38$, $p = 0.02$, see Fig 3B). No significant heterogeneity between the studies was found ($Q(1) = 0.66$, $p = 0.42$). This test should be interpreted with caution however, as its power to detect heterogeneity depends on the number of studies.

We also ran a second fixed effect meta-analysis to integrate results of dlPFC stimulation on enhancement. Over the two studies combined, dlPFC stimulation did not influence enhancement ($g = -0.19$ [−0.58, 0.21], $Z = -0.93$, $p = 0.35$). No significant heterogeneity between the studies was found ($Q(1) = 0.07$, $p = 0.80$).

To test for differences in participant characteristics between our study and the earlier one [7], we compared memory performance under vertex stimulation for to-be-remembered lists (R1 and R2) between studies. Participants in [7] showed significantly higher memory performance in general ($M = 57.7\%$, $SD = 24.9\%$) than participants in our study ($M = 39.4\%$,

$SD = 23.3\%$, $t(42.97) = 2.58$, $p = 0.01$, $g = -0.75$ [$-1.34$, $-0.16$]). Age-wise, participants in our study were on average three years younger ($M = 19.2a$, $SD = 2.0a$) than participants in the study by Hanslmayr et al. ($M = 22.2a$, $SD = 2.3a$, $Z = 5.45$, $p < 0.0001$). Concerning gender, participants in their sample [7] were more evenly distributed (18 male, 26 female) than our participants (4 male, 20 female), but not to a significant amount ($\chi^2 = 4.17$, $p = 0.06$).

## Discussion

In this study, we attempted to solidify evidence for a causal role of the dlPFC in directed forgetting by replicating an existing rTMS directed forgetting study [7]. Furthermore, we attempted to elucidate the neuronal effects of our dlPFC stimulation parameters by testing for potential aftereffects on a Stroop task.

We were able to run a functioning directed forgetting paradigm, as can be seen in our manipulation check: Instructions to forget list 1 led to both decreased memory for list 1 words (forgetting) and increased memory for list 2 words (enhancement) compared to instructions to remember list 1—the standard and often-found results of forget cues. Contrary to the study our experiment was attempting to replicate [7] and to our hypothesis, we did not replicate the finding that left dlPFC stimulation using 1Hz rTMS boosts forgetting. However, as can be seen by the small $BF_{01}$, our results also did not present substantial evidence against such an effect. As we found no significant increase of forgetting, but also no evidence against such an increase, our results on their own should be interpreted as inconclusive. Recently developed guidelines recommend a standardized integration of the results of original studies with replication attempts using a continuously cumulating meta-analysis [24]. This meta-analytic integration should be interpreted with the understanding that our replication was only conducted due to an earlier study [7] finding a significant result in the first place. This first-study-effect potentially inflates combined effect sizes in meta-analyses on small study sets [28]. Nevertheless, the two studies show a significant increase of voluntary forgetting due to dlPFC stimulation in combination. As the lower 95% confidence interval border of the combined effect size still stands close to zero, further replication attempts and subsequent updating of the meta-analysis is recommended to reliably demonstrate this causal role of the dlPFC. As both left and right dlPFC have been implicated in active memory control paradigms [3], future stimulation studies should also investigate the role of right dlPFC in directed forgetting and think/no-think tasks.

While our within-subject design eliminates between-subject confounds due to lesion or stimulation site, one general rTMS confound remains: More frontal stimulation locations (e.g. over the dlPFC) induce higher amounts of facial twitching compared to vertex stimulation. We asked our participants whether they felt distracted by the twitching caused by dlPFC stimulation during the debriefing, which most of them denied. The fact that stimulating dlPFC using tDCS also boosts forgetting [16] can be interpreted as evidence that forgetting modulations are induced by the stimulation itself, not by its side-effects on muscle twitches.

What might be the reason for this non-replication? While we cannot rule out the possibility that the original study found a false-positive result, the worse memory performance of our participants compared to the ones tested in [7] might have limited our ability to detect increased forgetting. In addition, age differences between the studies might have limited the comparability of the effects of rTMS stimulation: As the prefrontal cortex does not mature until the third decade of life [12], differences in dlPFC structure and baseline activity between our participants and the ones in the to-be-replicated study [7] due to a difference in age might have led to differential effects of the stimulation [29, 30].

Results of our Stroop task fit with a pulse-by-pulse effect of our rTMS parameters, free from stimulation aftereffects. As neither the incongruency effect, nor the negative priming effect, nor the overall pattern of reaction times were modulated after dlPFC compared to after vertex stimulation, we found no evidence of any post-stimulation aftereffects of the rTMS stimulation on inhibitory processing. This fits our assumption that each TMS pulse had immediate effects on neuronal activity in the dlPFC under our parameters without inducing strong long-term cognitive effects. Nevertheless, the fact that rTMS-induced changes in the forgetting effect and in the Stroop incongruency effect were weakly correlated over subjects might hint at weak persistent effects on dlPFC functioning, as have been described before for left dlPFC stimulation [31].

Our study provides important results on the viability of repeated list-method directed forgetting paradigms. As one of the first studies to run a second directed forgetting task after asking participants to recall to-be-forgotten items, we found no evidence that this repetition eliminates forgetting in the second directed forgetting task. This opens up the possibility to investigate directed forgetting in within-participant designs. In addition to these results, none of our participants reported that they ignored the forget cue after being asked to recall to-be-forgotten words for the first time.

Our results stand in disagreement with rTMS [7] and tDCS studies [16] that both found modulated forgetting due to dlPFC stimulation. Further dlPFC stimulation studies will be necessary to reliably establish a causal role of the dlPFC in directed forgetting. Our finding that 45s of 1Hz rTMS did not induce aftereffects on inhibitory functioning fit findings in cats, which showed that a minimum of five minutes of 1Hz rTMS is necessary to induce long-term depression of neuronal firing rates [32].

Given that the meta-analytic combination of both studies testing for effects of dlPFC stimulation on forgetting produces a significant effect, and given that this effect fits the fact that tDCS stimulation of the prefrontal cortex abolishes directed forgetting [16], we understand the current sum of evidence to be in favor of a causal role of the dlPFC in directed forgetting. As conclusively finding an active forgetting mechanism would have wide-ranging cognitive and clinical implications, we hope that future studies will further test the causality of the regularly found dlPFC activation in directed forgetting, for example by attempting additional replications of this paradigm.

## Author Contributions

**Conceptualization:** Verena Braun, Simon Hanslmayr.

**Data curation:** Benjamin J. Stauch, Verena Braun.

**Formal analysis:** Benjamin J. Stauch, Verena Braun, Simon Hanslmayr.

**Funding acquisition:** Simon Hanslmayr.

**Investigation:** Benjamin J. Stauch, Verena Braun.

**Methodology:** Verena Braun, Simon Hanslmayr.

**Project administration:** Verena Braun, Simon Hanslmayr.

**Resources:** Simon Hanslmayr.

**Software:** Benjamin J. Stauch, Verena Braun.

**Supervision:** Verena Braun, Simon Hanslmayr.

**Visualization:** Benjamin J. Stauch.

**Writing – original draft:** Benjamin J. Stauch.

**Writing – review & editing:** Benjamin J. Stauch, Verena Braun, Simon Hanslmayr.

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
