## [Decision Letter · Decision Letter 0]

1 Jun 2020

PONE-D-20-12383

Probing the causal involvement of dlPFC in directed forgetting using rTMS - A replication study

PLOS ONE

Dear Dr. Hanslmayr,

Thank you for submitting your manuscript to PLOS ONE. After careful consideration, we feel that it has merit but does not fully meet PLOS ONE’s publication criteria as it currently stands. Therefore, we invite you to submit a revised version of the manuscript that addresses the points raised during the review process.

Both reviewers appreciate the clarity and openness of design and analysis, and I agree.

Reviewer 2 has only minor remarks, Reviewer 1 has more fundamental ones. These are generally very relevant, and I would like to you to address them. However, I do not agree with the last comment of reviewer 1 about the meta-analysis. I do think it is of interest, and would thus like to ask you to keep it in the paper.

Moreover, I suggest to group all analyses of the current data set, and thus to move the order interaction (Effects of running…) up in the text (before the meta-analysis). The authors could also consider to plot the order effect.

In addition, it could help to show a high-level picture showing the two stimulation conditions in Figure 1. Also, in Figure 1B, instead of dots, why not just show two tasks?

Minor remark

- Does the r in rTMS stand for repetitive or rhythmic?

We look forward to receiving your revised manuscript.

Kind regards,

Tom Verguts

Academic Editor

PLOS ONE

Journal Requirements:

Reviewers' comments:

Reviewer's Responses to Questions

**Comments to the Author**

1. Is the manuscript technically sound, and do the data support the conclusions?

Reviewer #1: Partly

Reviewer #2: Yes

2. Has the statistical analysis been performed appropriately and rigorously? 

Reviewer #1: Yes

Reviewer #2: Yes

3. Have the authors made all data underlying the findings in their manuscript fully available?

Reviewer #1: Yes

Reviewer #2: Yes

4. Is the manuscript presented in an intelligible fashion and written in standard English?

Reviewer #1: Yes

Reviewer #2: Yes

5. Review Comments to the Author

Reviewer #1: Brief Summary

The manuscript reports one experiment aimed to replicate a previous study that found that rTMS over the left DLPFC enhanced list-method directed forgetting. The specific goal here was to replicate this effect with a within-participant (DLPFC vs vertex) design to overcome the potential drawbacks of a between-group manipulation. The basic directed forgetting effect was found. However, rTMS of the DLPFC did not modulate forgetting. In addition, the authors explored if the rTMS parameters used altered incongruency and negative priming effects in a Stroop task performed right after stimulation. Since these effects were also not modulated by rTMS, they suggest that stimulation did not produce a virtual lesion at the DLPFC.

General Evaluation and comments

Overall, I enjoyed reading this manuscript. I do appreciate that it reports a preregistered experiment aimed to replicate findings from the same laboratory. This is a valuable and gritty approach that would be very much desirable to take in science. Also, the experiment addresses a very relevant and timely research question (the causal role of the left DLPFC in memory regulation). That said, I have some serious concerns about the manuscript and the experiment that prevent me from recommending its publication at the moment. I will detail them below.

1. A better (theoretically grounded) justification of the rationale for stimulating the left DLPFC should be provided in the introduction (beyond mere replication). The authors use the findings of a number of brain-related studies to point to the role of the DLPFC in directed forgetting. However, some of these studies found that it is the right DLPFC that seems to be more involved in active forgetting. In fact, a very interesting paper co-authored by one of the authors of the reviewed manuscript (Anderson and Hanslmayr, 2014) supports the idea that motivated forgetting (as measured with different experimental paradigms) recruits predominantly right-lateralized PFC regions (with the DLPFC and the VLPFC being core ones) in charge of downregulating memories. Hence, why should we expect stimulation of the left DLPFC to affect forgetting that is interpreted by the authors (and myself) as an aftereffect of inhibitory-like control over memory?

2. My main concern relates to the (memory) testing procedure. In this case (but also in the previous work), the authors used a free recall test by asking participants to recall as many items as possible from List 1 (R or F) and List 2 (always R). Hence, there was no explicit instruction to recall first either List 1 or List 2 items. In my opinion, this procedure overlooks the role that output order may play in list-method DF procedures, which makes the interpretation of the findings regarding both lists problematic (see Pastötter, Kliegl and Bäuml 2012; M&C). While I think the analyses on the enhancement effect could simply be removed from the paper (I do not see this effect so relevant here), I do encourage the authors to appropriately address the issue that concerns the forgetting effect. At least, they should demonstrate that output interference is not significantly affecting List 1 recall or that it similarly affects R1 and F1. In any case, I think this is a clear limitation of the study that has to be overtly recognized. Moreover, I am afraid the forgetting effect in this experiment could have been maximized because of the within-participant design used and many and many lists to be learned (interference from previous lists?). Hence, analyses on intrusions could maybe help.

3. I cannot follow the rationale for using a Stroop/priming negative task (and its behavioral indexes) to elucidate the neuronal effects of the DLPFC stimulation. I understand this could be informative, but it is still a very indirect way of obtaining evidence to draw conclusions on neural activity. At the same time, I also think that failing to observe changes in performance for incongruent trials and the modulation of the negative priming effect does not necessarily imply that stimulating the left DLPFC did not cause a virtual lesion. It is matter of debate the specific role that this region plays in Stroop-like tasks (i.e., the review by Xu et al. 2016 showed that interference/inhibitory control in non-emotional tasks also recruits the right inferior PFC). I wonder why the authors did not totally replicate their previous experiment by also recording EEG. This would have been the right way to examine the neuronal effect of their stimulation manipulation.

4. In my opinion, the meta-analysis (with only 2 studies) performed is not appropiate and may be misleading.

Reviewer #2: This is a neat study with humble goal but solid methodology. Findings are not really conclusive but points to the need for further confirmation of the causal role of BA9 in directed forgetting.

My only 2 comments are:

- given the absence of evidence for any effect of the 1 Hz rTMS design on BA9 function, other interferential techniques could be considered in the future (e.g. Viejo-Sobera et al. 2017, which should be cited).

- if rTMS effect were present but too weak to come out significantly, we would still expect TMS impact on directed forgetting and Stroop performance to correlate with each other. This could be tested.

6. PLOS authors have the option to publish the peer review history of their article (what does this mean?). If published, this will include your full peer review and any attached files.

Reviewer #1: No

Reviewer #2: No

---

## [Author Response · Author response to Decision Letter 0]

17 Jun 2020

Both reviewers appreciate the clarity and openness of design and analysis, and I agree.

Reviewer 2 has only minor remarks, Reviewer 1 has more fundamental ones. These are generally very relevant, and I would like to you to address them. However, I do not agree with the last comment of reviewer 1 about the meta-analysis. I do think it is of interest, and would thus like to ask you to keep it in the paper.

Moreover, I suggest to group all analyses of the current data set, and thus to move the order interaction (Effects of running…) up in the text (before the meta-analysis). The authors could also consider to plot the order effect.

We agree with this suggestion and have altered the text order and added the suggested plot to Figure 3.

In addition, it could help to show a high-level picture showing the two stimulation conditions in Figure 1. Also, in Figure 1B, instead of dots, why not just show two tasks?

We think that these are excellent suggestions. A high-level picture showing the approximate location of both stimulation locations on an example brain surface has been added to Figure 1. Figure 1B has been modified as suggested.

Minor remark

- Does the r in rTMS stand for repetitive or rhythmic?

We missed to properly define this abbreviation. Line 41 has been amended to “The repetitive transcranial magnetic stimulation (rTMS) study we are attempting to replicate stimulated left dlPFC (specifically BA9) after the forget or remember instruction using 45 seconds of 1Hz rTMS.”

Reviewer #1: Brief Summary

The manuscript reports one experiment aimed to replicate a previous study that found that rTMS over the left DLPFC enhanced list-method directed forgetting. The specific goal here was to replicate this effect with a within-participant (DLPFC vs vertex) design to overcome the potential drawbacks of a between-group manipulation. The basic directed forgetting effect was found. However, rTMS of the DLPFC did not modulate forgetting. In addition, the authors explored if the rTMS parameters used altered incongruency and negative priming effects in a Stroop task performed right after stimulation. Since these effects were also not modulated by rTMS, they suggest that stimulation did not produce a virtual lesion at the DLPFC.

General Evaluation and comments

Overall, I enjoyed reading this manuscript. I do appreciate that it reports a preregistered experiment aimed to replicate findings from the same laboratory. This is a valuable and gritty approach that would be very much desirable to take in science. Also, the experiment addresses a very relevant and timely research question (the causal role of the left DLPFC in memory regulation). That said, I have some serious concerns about the manuscript and the experiment that prevent me from recommending its publication at the moment. I will detail them below.

1. A better (theoretically grounded) justification of the rationale for stimulating the left DLPFC should be provided in the introduction (beyond mere replication). The authors use the findings of a number of brain-related studies to point to the role of the DLPFC in directed forgetting. However, some of these studies found that it is the right DLPFC that seems to be more involved in active forgetting. In fact, a very interesting paper co-authored by one of the authors of the reviewed manuscript (Anderson and Hanslmayr, 2014) supports the idea that motivated forgetting (as measured with different experimental paradigms) recruits predominantly right-lateralized PFC regions (with the DLPFC and the VLPFC being core ones) in charge of downregulating memories. Hence, why should we expect stimulation of the left DLPFC to affect forgetting that is interpreted by the authors (and myself) as an aftereffect of inhibitory-like control over memory?

We agree with the reviewer that the right/bilateral dlPFC has been consistently found to be activated in active memory control paradigms (think/no-think tasks and item-method directed forgetting paradigms). However, there has not yet been conclusive evidence for complete lateralization to either side, and the only study we are aware of combining list-wise directed forgetting tasks and fMRI found activation in left dlPFC (Hanslmayr et al., 2012). As has been argued before, there are probably significant cognitive differences between item-method and list-method directed forgetting paradigms, because the remember/forget-cue is shown directly during encoding in item-method tasks and later in list-method tasks. As activations in both hemispheres have been found, and as it is as of yet unclear how lateralization depends on the task, we chose to target the brain coordinates most strongly activated when a task very similar to ours was used. We do agree, however, that future work should also target right dlPFC, ideally under a set of diverse active memory control tasks. We appreciate that this issue was not clearly expressed in our manuscript and have therefore added text in the introduction and discussion which hopefully clarifies this point.

2. My main concern relates to the (memory) testing procedure. In this case (but also in the previous work), the authors used a free recall test by asking participants to recall as many items as possible from List 1 (R or F) and List 2 (always R). Hence, there was no explicit instruction to recall first either List 1 or List 2 items. In my opinion, this procedure overlooks the role that output order may play in list-method DF procedures, which makes the interpretation of the findings regarding both lists problematic (see Pastötter, Kliegl and Bäuml 2012; M&C). While I think the analyses on the enhancement effect could simply be removed from the paper (I do not see this effect so relevant here), I do encourage the authors to appropriately address the issue that concerns the forgetting effect. At least, they should demonstrate that output interference is not significantly affecting List 1 recall or that it similarly affects R1 and F1. In any case, I think this is a clear limitation of the study that has to be overtly recognized. Moreover, I am afraid the forgetting effect in this experiment could have been maximized because of the within-participant design used and many and many lists to be learned (interference from previous lists?). Hence, analyses on intrusions could maybe help.

We agree with the reviewer in that output order can have an effect on directed forgetting effects in the list method. We are aware of the study by Pastoetter and colleagues who indeed demonstrated that the enhancement effect depends on list output order: It is diminished if subjects are instructed to recall words form list 1 first. Of note, this same paper also shows also that list-1 forgetting is not affected by output order. With respect to our study, these results suggest that any effect of rTMS on enhancement could be masked if dlPFC stimulation affected which list the subjects attempted to recall first. This was, however, not the case: While subjects were more likely to start recall with list 1, when previously instructed to remember list 1 than when previously instructed to forget it (p < 0.05 for recalled words 1-4), stimulation site did not influence recall order (all p > 0.05 for recalled words 1-5) nor did it moderate the difference in recall order between the remember and forget condition (all ps > 0.05 for words 1-5). We have added this control analysis to Methods, Discussion and the code repository. Unfortunately, intrusions were not recorded and hence are not available for analysis.

3. I cannot follow the rationale for using a Stroop/priming negative task (and its behavioral indexes) to elucidate the neuronal effects of the DLPFC stimulation. I understand this could be informative, but it is still a very indirect way of obtaining evidence to draw conclusions on neural activity. At the same time, I also think that failing to observe changes in performance for incongruent trials and the modulation of the negative priming effect does not necessarily imply that stimulating the left DLPFC did not cause a virtual lesion. It is matter of debate the specific role that this region plays in Stroop-like tasks (i.e., the review by Xu et al. 2016 showed that interference/inhibitory control in non-emotional tasks also recruits the right inferior PFC). I wonder why the authors did not totally replicate their previous experiment by also recording EEG. This would have been the right way to examine the neuronal effect of their stimulation manipulation.

We agree with the reviewer that the conducted Stroop/negative priming tasks do not elucidate the neuronal effects of the applied rTMS stimulation and realise that our motivation for running this task was worded in a suboptimal manner. We were not directly interested in the neuronal effects of our stimulation and are of the opinion that these cannot be properly investigated in studies utilizing non-invasive measures in human subjects. Instead, we included this task to check for stimulation aftereffects on the same inhibitory processes that created the increased forgetting effect in the study we were attempting to replicate. Our wording has been modified to reflect this focus on inhibitory processing. Comments regarding the underlying changes in neuronal processes have been removed.

4. In my opinion, the meta-analysis (with only 2 studies) performed is not appropiate and may be misleading.

We agree with the reviewer that meta-analyses with low study numbers should be interpreted with caution, and stress that point in our discussion (“This meta-analytic integration should be interpreted with the understanding that our replication was only conducted due to an earlier study finding a significant result in the first place. This first-study-effect potentially inflates combined effect sizes in meta-analyses on small study sets.”).

As can be seen in Figure 3b however, the estimated effect confidence intervals of the original study and our replication largely overlap. It would therefore, in our view, be misleading to dichotomize and present their results as strictly opposite to each other. As shown in Braver, Thoemmes & Rosenthal (2014), the continuously cumulating meta-analytic (CCMA) approach yields a more accurate picture of the combined results of several replication studies, even for very low study numbers. In fact, their paper explicitly recommends the CCMA approach for one-study replications of existing significant results. We therefore are of the opinion that a CCMA is the appropriate way to combine our replication attempt with the original study. 

Reviewer #2: This is a neat study with humble goal but solid methodology. Findings are not really conclusive but points to the need for further confirmation of the causal role of BA9 in directed forgetting.

My only 2 comments are:

- given the absence of evidence for any effect of the 1 Hz rTMS design on BA9 function, other interferential techniques could be considered in the future (e.g. Viejo-Sobera et al. 2017, which should be cited).

We thank the reviewer for the relevant citation on the effects of rTMS of left dlPFC on Stroop performance, which we have incorporated into the discussion.

We are unsure which precise inferential techniques the reviewer is referring to here. In the cited paper, the authors write: “In order to rule out the possibility of a type II error we conducted a nonparametric K-sample test on the equality of medians”. To our understanding, such a test is a nonparametric alternative of a t-test. We have computed non-parametric paired Wilcoxon signed rank tests for all three non-significant t-tests that compared Stroop effects between stimulation conditions. Like the t-tests, they were non-significant (p = 0.55, p = 0.53 and p = 0.94, for the incongruency, incongruency with neutral trials and negative priming effects, respectively). Because differences of reaction times are generally assumed to be normally distributed in the population, we think it to be appropriate to report t-tests of these differences in the paper itself.

- if rTMS effect were present but too weak to come out significantly, we would still expect TMS impact on directed forgetting and Stroop performance to correlate with each other. This could be tested.

We thank the reviewer for this excellent suggestion. We have correlated rTMS effects on directed forgetting and Stroop effects, and have indeed found a weakly significant correlation between changes in the forgetting effect and changes in the incongruency effect. We agree that this might hint at weak persistent effects on inhibitory functioning in general, and have updated the Results, Discussion and code repository to reflect this analysis.

---

## [Editor Report · Decision Letter 1]

6 Jul 2020

Probing the causal involvement of dlPFC in directed forgetting using rTMS - A replication study

PONE-D-20-12383R1

Dear Dr. Hanslmayr,

We’re pleased to inform you that your manuscript has been judged scientifically suitable for publication and will be formally accepted for publication once it meets all outstanding technical requirements.

Kind regards,

Tom Verguts

Academic Editor

PLOS ONE

Additional Editor Comments (optional):

small typo: page 3: in to  to
---

## [Editor Report · Acceptance letter]

8 Jul 2020

PONE-D-20-12383R1 

Probing the causal involvement of dlPFC in directed forgetting using rTMS - A replication study 

Dear Dr. Hanslmayr:

I'm pleased to inform you that your manuscript has been deemed suitable for publication in PLOS ONE. Congratulations! Your manuscript is now with our production department. 

Kind regards, 

on behalf of

Dr. Tom Verguts 

Academic Editor

PLOS ONE